# 2D Pose Estimation vs. Inertial Measurement Unit-Based Motion Capture in Ergonomics: Assessing Postural Risk in Dental Assistants

**DOI:** 10.3390/bioengineering12040403

**Published:** 2025-04-10

**Authors:** Steven Simon, Jonna Meining, Laura Laurendi, Thorsten Berkefeld, Jonas Dully, Carlo Dindorf, Michael Fröhlich

**Affiliations:** Department of Sports Science, RPTU University Kaiserslautern-Landau, 67663 Kaiserslautern, Germany; jomeining@aol.de (J.M.); laulaurendi@gmail.com (L.L.); thorsten.berkefeld@rptu.de (T.B.); jonas.dully@rptu.de (J.D.); carlo.dindorf@rptu.de (C.D.); michael.froehlich@rptu.de (M.F.)

**Keywords:** ergonomics, motion capture, pose detection, work-related discomfort, dental assistant health, observational methods

## Abstract

The dental profession has a high prevalence of musculoskeletal disorders because daily working life is characterized by many monotonous and one-sided physical exertions. Inertial measurement unit (IMU)-based motion capture (MoCap) is increasingly utilized for assessing workplace postural risk. However, practical alternatives are needed because it is time-consuming and relatively cost intensive for ergonomists. This study compared two measurement technologies: IMU-based MoCap and a time-effective alternative, two-dimensional (2D) pose estimation. Forty-five dental assistant students (all female) were included (age: 19.56 ± 5.91 years; height: 165.00 ± 6.35 cm; weight: 63.41 ± 13.87 kg; BMI: 21.56 ± 4.63 kg/m^2^). A 30 s IMU-based MoCap and image-based pose estimation in the sagittal and frontal planes were performed during a representative experimental task. Data were analyzed using Cohen’s weighted kappa and Bland–Altman plots. There was a significant moderate agreement between the Rapid Upper Limb Assessment (RULA) score from IMU-based MoCap and pose estimation (κ = 0.461, *pB* = 0.006), but no significant poor agreement (*p* > 0.05) regarding the body regions of the upper arm, lower arm, wrist, neck, and trunk. These findings indicate that IMU-based MoCap and pose estimation moderately align when assessing the overall RULA score but not for specific body parts. While pose estimation might be useful for quick general posture assessment, it may not be reliable for evaluating joint-level differences, especially in body areas such as the upper extremities. Future research should focus on refining video-based pose estimation for real-time postural risk assessment in the workplace.

## 1. Introduction

The dental profession involves significant physical exertion and a prolonged nature of work, rendering dental professionals particularly susceptible to musculoskeletal disorders (MSDs) [1,2]. The prevalence of MSDs among dentists is notably higher than that in the general population, with the neck, shoulders, and back being the most commonly affected regions [1,3]. This issue is especially pronounced among dental, dental hygiene, and dental assistant students, who spend extended hours in pre-clinical laboratories practicing on phantom heads [2]. Various factors contribute to poor posture in this context, including the need to coordinate movements with assistants for a smooth workflow, to contort their bodies for an optimal intraoral view, and to ensure patient comfort, which often requires adopting non-ergonomic positions [2,4]. Continuous strain on the joints in one direction may cause muscle imbalances or structural tissue damage, ultimately leading to lower back pain [2,5]. Dental hygienists and assistants, normally positioned on the left side of the patient, assume responsibility for suction and holding tasks, which are typically performed while seated. They are exposed to high postural loads owing to the large amount of static and holding work [6]. Therefore, ergonomic work plays a crucial role in their professional life [2,4]. In particular, students have their entire professional lives ahead, indicating that significant health benefits may be created through guided training at a young professional age. Furthermore, institutions are expected to intervene by implementing ergonomic practices to benefit their students [2].

Observational methods are commonly used to assess postural load and provide MSD risk scores [7,8]. Tools such as the Rapid Entire Body Assessment, Rapid Upper Limb Assessment (RULA), and Ovako Work Posture Assessment System are widely applied in ergonomics. Among these, RULA is particularly suitable for evaluating postural strain [9]. However, it is limited by the fact that observational assessment procedures not only require the involvement of a field expert for labor-intensive manual analysis but also rely heavily on the subjective judgment of the evaluator, potentially resulting in substantial variability among raters [10]. As one of the most frequently used observational methods, RULA can be applied to data collection using an inertial measurement unit (IMU) [8,11] and has proven advantageous for recording postural load in the work process [12]. Maurer-Grubinger et al. [6] highlighted the benefits of IMU-based assessments by comparing different dental work routines and demonstrated their potential for detailed ergonomic analyses at multiple RULA levels. Holzgreve et al. [13] showed that the ergonomic risk in all dental disciplines involved moderate to high RULA scores. The application of IMU has several advantages; however, it presents considerable costs and measurement efforts to ergonomists. Camera-based methods do not require markers or interfere with subjects, making them a crucial approach for the future advancement of ergonomic assessments [14]. Nowara et al. [15] assessed deviations between the conventional paper-and-pencil method of RULA risk assessment supported by images from a tablet, and a kinematic data-based approach. The comparison was conducted using situational recordings of a dental team working in a dental treatment setting. The results suggested that the IMU-based RULA offers a more precise and reliable risk assessment, overcoming the observer-related inconsistencies and limitations of static image-based evaluations. However, the paper-and-pencil method of RULA with a tablet remains a cost-effective and accessible method, particularly for static postures, although its tendency to yield lower risk scores should be considered, and 26.83% of assessments show errors (e.g., due to a limited knowledge of the observational method) [15,16].

A potential alternative cost- and time-effective tool is two-dimensional (2D) human pose estimation using mobile devices [17,18]. Pose estimation can be defined as “the process of automatically tracking and determining the body’s anatomical landmarks, body segments, or joint locations in video images using machine learning, enabling the quantification of human movement without marker and sensor attachments to the human body” [19]. One of the key advantages of pose estimation is its ease of use [20]. These algorithms require only video footage or images as inputs to track and analyze body postures. Unlike marker- or IMU-based motion capture (MoCap), specialized cameras such as depth cameras and the complex instrumentation of the subject are not necessary. This provides significant benefits in terms of cost and time efficiency, allowing movement to be recorded and analyzed outside of a laboratory setting [20]. The model MoveNet has the potential to classify postures in a work environment [21]. However, further research on pose estimation is required [21] to assess whether its ergonomic risk assessment achieves results comparable to those of IMU-based MoCap.

To address these research gaps, this study utilized two technologies for assessing postural risk in the workplace: IMU-based MoCap and 2D human pose estimation as a time- and cost-effective alternative for practical ergonomists. It was hypothesized that the IMU-based MoCap and 2D human pose estimation RULA scores would be positively correlated.

## 2. Materials and Methods

### 2.1. Participants

A total of 45 dental assistant students (all female) from Worms BBS school (Germany) voluntarily participated in this study (age: 19.60 ± 5.98 years; height: 165.08 ± 6.41 cm; weight: 62.70 ± 13.20 kg; body mass index: 21.39 ± 4.54 kg/m^2^; percentage of right-handed participants: 91.1%) (see Table 1).

Professional dental assistants are characterized as follows [22]:Usually remain in the same position for long periods because of the monotonous work (holding and suctioning).Experience frequent long periods in a chair without a break because of patient preparation and follow-up (e.g., removal of temporaries and impressions).Sitting position is subordinate to the positioning of the dentist.Frequently experience a poor field of vision (small intraoral view, intricate working area such as for filling application, and the dentist takes priority for the best view while the dental assistant must adapt).Additional equipment is required for compensation (e.g., magnifying or prism glasses or armrests on chairs).

Each participant was informed verbally and in writing, and signed an informed consent form regarding data rights, recording videos, and participation in the study procedures. The study was conducted in accordance with the guidelines of the Declaration of Helsinki and approved by the institutional ethics committee (Ethikkommission RPTU Kaiserslautern-Landau, No. 66-2023). The inclusion criterion was defined as a full-time dental assistant student with a minimum of 3 months of professional experience. The exclusion criteria comprised current injuries to the musculoskeletal system, acute restriction of physical activity, and surgical treatment of the musculoskeletal system in the previous 4 weeks. Participants were asked for their medical history (musculoskeletal impairments), but none of them reported a spinal pathology.

### 2.2. Experimental Settings

Figure 1 shows the experimental setup in the dental laboratory in which the working posture was measured.

A webcam (Logitech, Vaud, Switzerland) and two tablets in the sagittal and frontal planes (iPad, Apple, Cupertino, CA, USA), as well as the Awinda station (Movella), were positioned approximately 2.5 m from the patient to guarantee good video quality and an optimal view (Figure 2). In contrast to Noewara et al. [15], who recorded from the front, top, and rear sides, the tablets were positioned in the frontal and sagittal planes at a standardized height (88 cm) and distance (2.52 m) to the experimental setup to ensure a good view of the entire participant’s body and to minimize parallax errors in 2D images. All recordings were performed by the same two ergonomists who were educated in health sciences.

### 2.3. Experimental Task

All dental assistant students performed the same experimental tasks. The same test supervisor read the task to each participant before measurements began.

Sit in the assistant chair.Place a filling occlusally on tooth 36.Use the large suction cup to hold the lingual and the mirror so that the dentist has a clear view of the affected tooth surface.Assume a position as comfortable as possible for the patient and yourself.Remain in this position during the acoustic signal.You may try out the position once.Finally, perform the task.

This represents a typical dental assistant task that is performed frequently during daily working life. Dental assistants generally adopt many static postures and hold tools in the patient’s mouth for the dentist. Therefore, static analysis can be used to depict a typical working posture used frequently in the work process. Data for the experimental task were recorded on two measurement dates. This was due to the availability of the participants. Measurements were taken between September and December 2024 at the same time of day (9 am to 2 pm). The holding and guide arms were swapped at the first and second measurements to include both sides of the work activity with the 2D pose estimation method (ToM_1_ = mirror in right hand, suction cup in left hand; ToM_2_ = mirror in left hand, suction cup in right hand).

### 2.4. Pose Estimation and IMU-Based MoCap

Two images were captured from two perspectives (frontal and sagittal) during the experimental task using a tablet (Apple iPad120 FPS, 1080P). For 2D posture analysis, the Ergofreude application (lebensfreude Gesundheitsmanagement, Saarbrücken, Germany), which is based on motion estimation in MoveNet (Google Inc., Mountain View, California USA) via mobile device applications, was used. Human pose estimation is a promising technology for measuring joint angles and facilitating semiautomatic ergonomic postural assessments in real work environments [10]. MoveNet was designed to identify 17 key points in the human body [23] and is applied in corporate health management in the German health industry and is therefore of relevant industrial interest for further developing the practical field of ergonomics. A comparative analysis of Jo et al. [24] showed that MoveNet was the fastest model in comparison to OpenPose and PoseNet. As a bottom-up model, it relies on the feature extractor MobileNet V2 and the TensorFlow Object Detection Application Programming Interface [24]. MoveNet achieves accurate results with image data and has significant potential for ergonomic assessments [21]. A person is identified when the regression aligns with the arrangement of the prepared keypoints. Each pixel is multiplied by a weight inversely correlated with the distance from the regressed keypoint [24]. MoveNet uses MobileNet V2 as a feature extractor and is enhanced using a Feature Pyramid Network. The feature extractor is connected to four key prediction heads [23] (Figure 2):Person Center Heatmap: Detects the geometric centers of individuals.Keypoint Regression Field: Predicts a full set of keypoints for each person.Person Keypoint Heatmap: Locates keypoints independently of person instances.Two-Dimensional Per-Keypoint Offset Field: Computes local offsets for subpixel keypoint precision.

All models were trained using the TensorFlow Object Detection Application Programming Interface [23]. The obtained joint angles were evaluated using the RULA ergonomic evaluation scheme [7].

The participants underwent a 30 s MoCap using an IMU (Xsens Technologies B.V., Enschede, The Netherlands) (Figure 2). The IMU was calibrated before each measurement in N-pose and comprised a three-axis accelerometer (±16 g), a three-axis gyroscope (±2000° per second), and a three-axis magnetometer (±1.9 Gauss) [25]. This represents a robust and precise reference system for reconstructing three-dimensional motion in the workplace [26] and can deliver repeatable and accurate ergonomic risk scores [8,27].

Score A evaluated the arms and wrists, considering muscle activity (repetitive movements or static postures held for more than 1 min) and external forces (<2 kg, 2–10 kg, and >10 kg, repetitive or static). Similarly, score B assessed the neck, trunk, and legs by incorporating the same muscle activity and force parameters. The final score (C) was derived from scores A and B, representing the overall MSD risk level on a scale of 1 to 7. A score of 1 indicates a low risk of work-related MSD, whereas scores of 3 or 4 suggest the potential need for ergonomic intervention or procedural adjustments. A score of 5 or 6 indicates an imminent necessity for modifications, whereas a score of 7 signifies a critical need for immediate changes in work procedures [28].

### 2.5. Data Processing

Kinematic data from the IMU-based MoCap (see Figure 3) were merged using MATLAB software (R2023a, The MathWorks, MA, USA), adapted to a self-written RULA evaluation scheme. Finally, the RULA score was determined from the relative time spent in each RULA score level (1–7) [13] (see Figure 4).

The angles detected via the Ergofreude application from the frontal and sagittal planes (Figure 3) were used to assess each evaluation criterion of the RULA, resulting in individual body part scores and a final score (see Table 2). All transverse plane parameters (upper arm, wrist, neck, and trunk twisting) were detected by the ergonomist who analyzed the pose estimation data.

The angle transfers and modifications for calculating the RULA score [29] based on pose estimation are listed in Table 2.

### 2.6. Data Analysis

All data were initially checked for normal distribution using the Shapiro–Wilk test in SPSS (version 29, SPSS Inc., Chicago, IL, USA). The Shapiro–Wilk test showed some violations of normality (pose estimation: RULA, trunk; IMU-based MoCap: RULA, neck, trunk, wrist, upper arm, and lower arm). Therefore, the median difference between the results was used for descriptive statistics.

Data visualization was performed using Microsoft Excel (version 16.78.3; Microsoft, Redmond, WA, USA) and JASP (version 0.19.0; JASP Team, Amsterdam, The Netherlands). Visual and expert-based analyses of the data were performed to detect outliers. Bland–Altman and scatter plots, including histograms, were used for data presentation.

Cohen’s weighted kappa was used to assess the reliability of the RULA scores of both technologies using SPSS and was interpreted according to Altman [30]. Given that final RULA scores range from 1 to 7, Cohen’s kappa [31] is an appropriate statistical measure for evaluating inter-method agreement while accounting for the degree of disagreement between categories. In addition, Mann–Whitney U tests were used to detect differences between the two technologies in the assessment of the RULA scores, and the effect size was determined by rank biserial correlation using JASP. Significant results were subjected to Bonferroni correction, and the significance level was *p* < 0.05.

A post hoc power analysis was conducted using G*Power 3.1 [32] to assess the achieved statistical power based on an effect size of r = 0.45, an alpha level of 0.05, and a total sample size of 45. The analysis indicated a statistical power of 0.91 (91.05%), suggesting that the study had a high probability of detecting a true effect.

## 3. Results

There was a highly significant moderate agreement κ = 0.461 ** (*p* < 0.001; Bonferroni-corrected *p*-value [*p_B_*] = 0.006) (95% confidence interval [CI], 0.281–0.642) between the RULA scores assessed by both technologies. There were no significant poor agreements regarding the body part scores of the upper arm (κ = 0.17, *p_B_* = 0.228; 95% CI, 0.007–0.339), lower arm (κ = 0.01, *p* = 0.87; 95% CI, −0.101–0.118), wrist (κ = 0.08, *p* = 0.26; 95% CI, −0.056–0.213), neck (κ = 0.10, *p* = 0.26; 95% CI, −0.037–0.240), and trunk (κ = 0.09, *p* = 0.12; 95% CI, −0.032–0.208).

Figure 5 presents the Bland–Altman plot of the RULA scores. All data were within the fluctuation range except for that of one participant. Bland–Altman plots of the body parts are presented in the Appendix A.

The mean (±standard deviation [SD]) overall RULA score for IMU-based MoCap and pose estimation was 4.82 ± 1.25 and 4.78 ± 0.97, respectively (95% CI, 0.281–0.642) (Table 3 and Figure 6). The results from both technologies are graphically presented in Figure 6 and Figure 7.

There were no significant differences in overall and some body part RULA scores between both technologies according to the Mann–Whitney U test results: overall (mean difference [Mean_Diff_] = 0.044, *p* = 0.977, rank biserial correlation = 0.004), neck (Mean_Diff_ = 0.201, *p* = 0.724, rank biserial correlation = −0.043), and upper arm (Mean_Diff_ = 0.274, *p* = 0.108, rank biserial correlation = 0.287). However, there were significant differences regarding the lower arm (Mean_Diff_ = 0.305, *p_B_* < 0.006, rank biserial correlation = 0.525), wrist (Mean_Diff_ = 0.381, *p_B_* < 0.006, rank biserial correlation = 0.453), and trunk (Mean_Diff_ = 0.476, *p_B_* = 0.006, rank biserial correlation = −0.497) (Figure 7).

## 4. Discussion

### 4.1. Main Findings

To the best of the authors’ knowledge, this is the first study to compare the measurement technologies of IMU-based MoCap and 2D pose estimation for the assessment of ergonomic risks for dental assistants using the RULA method. According to McAtamney [29], a RULA score exceeding 5 indicated that immediate investigation is required, in both methods. The mean RULA scores for IMU-based MoCap (4.82, SD ± 1.25) and pose estimation (4.78, SD ± 0.97) suggested that both technologies produced similar overall posture risk assessments. Inference statistical analysis revealed that IMU-based MoCap and pose estimation showed a significant moderate agreement (κ = 0.46, *pB* < 0.001). This suggests that the two methods produce similar results in the evaluation of overall posture risk. A slightly higher mean score for the IMU indicates a tendency for this technology to classify postural risk marginally more strictly than pose estimation. However, there are still notable discrepancies. The SD was greater for the IMU (1.25) than for pose estimation (0.97), suggesting that the IMU captures more variability in posture assessments, potentially owing to its sensitivity to subtle postural deviations. It must be considered that IMU scores represent a mean value from continuous kinematic data, whereas pose estimation represents two moment captures with only one angle estimation of the relevant joints. IMUs may offer deeper insights into ergonomic risks that pose estimation might not recognize. However, this variability could also stem from sensor drift, magnetic interference, or slight inconsistencies in sensor placement, affecting joint angle estimation. Consequently, these fluctuations may compromise the reliability of IMU-based postural risk assessments.

Poor-to-negligible agreement was observed for individual body part scores, indicating potential limitations in detecting specific postural deviations using one or both methods. The upper arm, lower arm, and wrist were scored slightly higher in the IMU than in pose estimation (Table 3), suggesting that the IMU detected slightly greater postural risks in the arm region. In contrast, pose estimation showed descriptively higher mean scores for the neck (3.43, SD ± 0.75) compared to the IMU (3.23, SD ± 1.32), but these differences were not significant. Of note, IMU exhibited a significantly larger SD (1.32 vs. 0.75), implying that it captured a broader range of neck postures and finer variations. The most notable difference was regarding trunk posture, where pose estimation recorded a higher mean score (2.43, SD ± 0.54) than the IMU (1.95, SD ± 0.35). This suggests that pose estimation should place greater emphasis on trunk deviations. This is not in accordance with a previous study [15] that reported comparable results in the B-score of RULA (neck, trunk, and leg analyses by IMU-based MoCap RULA) and the paper-and-pencil RULA method. A clear trend emerged when examining the variability (SD) across the body part scores: the IMU consistently had a higher SD than pose estimation, indicating that it can capture more detailed movement fluctuations, whereas pose estimation appears to produce less fluctuating postural estimates. The tracking errors in both technologies must be considered [33,34].

Alvarez [35] evaluated the accuracy of MoveNet in estimating joint angles and found that the average root mean square error (RMSE) was approximately 12°, whereas the implementation of a linear regression model reduced the RMSE to an average of 5°. However, the accuracy of MoveNet can be influenced by factors such as movement type, speed, and environmental conditions. Furthermore, owing to the dental setup, MoveNet had problems recognizing some of the joint marks (such as the wrists and hips) as reference values for trunk posture, comparable to the findings of another study [21]. In addition, because the dental assistant was sitting in front of the patient, measurements had to be repeated due to recognition of the patient’s body parts instead of those of the dental assistant. In OpenPose, another vision-based pose estimation method, Chen et al. [36] showed an accuracy of 97% in velocity, angle, and ratio. Kim et al. [10] reported that OpenPose has the potential to measure joint angles and perform semiautomatic ergonomic assessments; the overall total mean RMSE of the joint angles was 8.4°. Li et al. [17] estimated the RULA score from 2D human poses encoded with three-dimensional pose information, achieving a 93% accuracy in RULA action level classification. Furthermore, average joint center differences between 10 and 50 mm using multicamera markerless systems with OpenPose have been reported [37]. Therefore, this method might not offer the required accuracy to fully replace biomechanical applications at this point [19].

Regarding IMU-based MoCap, Teufl et al. [38] reported an RMSE below 2.40° for all joints using the Xsens IMU, whereas the mean range of motion error remained below 1.60°. Nevertheless, missing angle thresholds such as upper arm abduction or neck twisting must be considered, and IMUs can experience magnetic disturbances [12]. Algorithms that use IMU data to provide score-based results are an option, and proposals have recently been published [25]. MoCap with an IMU enables the recording of the kinematic data of work processes. Therefore, it is possible to determine the RULA score for each posture of the work process and allow a temporal distribution in the four risk areas, which represents the overall process of the chosen task. Consequently, a more objective determination of the total ergonomic load is feasible, leading to a more accurate assessment of workplace ergonomics [6]. Generally, this is an advantage over pose estimation because the observer cannot assess the entire occupational demand during a complete work shift [6].

These findings indicate that although pose estimation and IMU-based MoCap may yield similar overall RULA scores, their sensitivities to specific joint movements differ. The upper arm segment showed the highest agreement, although still poor, likely because both technologies can reliably detect gross arm movements. For industries with resource constraints, 2D pose estimation may serve as a viable, cost-effective alternative to IMU-based assessments, offering ergonomic evaluations without sacrificing accuracy in mean risk scores. Ergonomists must be trained on how the accuracy of pose estimation might vary depending on factors such as camera angles, lighting conditions, and environmental interference. A deeper understanding of these limitations can help them better interpret results and mitigate issues during assessments. However, assessment via pose estimation cannot accurately capture subtle joint angles in these regions. It is assumed that these two methods cannot be applied congruently. For tasks involving complex or highly dynamic postures, IMUs provide additional insights.

### 4.2. Methods

The participants were instructed to perform the same experimental task, but were not restricted in their behavior, and were allowed to set an individualized optimal arrangement of work instruments. This approach was deliberately selected to reflect the actual requirements of daily work. The experimental task replicates the phase of excavation and preparation. During the actual filling process, suction is ensured via the saliva ejector, while the dental assistant prepares the filling material and begins. Additionally, treating a tooth in position 16 or 26 would be significantly more challenging. Furthermore, some participants deviated from standard ergonomic guidelines when working at the chair—for example, their left arm did not rest against the torso while retracting.

The use of an IMU for recognizing employee exertion is increasingly becoming the focus of occupational science [8] because wearable IMUs are accurate, ergonomic risk assessment tools [27]. IMU-based MoCap relies on precise sensor placement [39], which can introduce variability if not positioned consistently across participants. Pose estimation relies on camera-based tracking, which may result in occluded body parts, perspective distortion, and depth perception inaccuracies. These dependencies can lead to variations in joint angle estimations and, consequently, impact reproducibility. Parallax errors occur in 2D video analysis when objects or body parts are not aligned correctly with the camera’s optical axis. This error arises because the 2D analysis lacks depth perception, leading to inaccuracies in measuring angles, distances, and movements [40]. This could explain why the lower arm and wrist movements, which are often partially obscured, show particularly low agreement.

Environmental dependencies play a crucial role in reproducibility. IMUs function independently of external lighting and camera positioning, making them more reliable across different settings. However, IMUs are susceptible to interference from ferromagnetic objects [41]. On the other hand, 2D pose estimation is sensitive to environmental factors, potentially limiting its consistency when applied in varied workplace conditions. Pose estimation can be influenced by the number, placement, and resolution of RGB cameras. The extent to which these factors affect the performance of RGB-based pose estimation has not been thoroughly investigated [10]. The attachment of the IMU makes it even more difficult to recognize the wrist position. The participants wore a tightly fitted Xsens upper-body suit because loose clothing may reduce the accuracy of the results. Only one side (right-hand side) was assessed, in contrast to the IMU, which enabled both sides to be evaluated and differentiated [6]. Another point of criticism arises when considering primarily the upper body. If the dental assistant’s seating position is set too high for the dentist, it becomes difficult to place their feet firmly on the ground and maintain an optimal thigh angle. This imbalance can lead to pelvic tilting, which in turn affects the torso. However, in this case, the core problem lies in the positioning of the legs.

However, both measurement technologies neglect the transverse plane. Although rotations cannot be detected with 2D human pose estimation, there are still limitations in the transverse plane when using an IMU. Because the rotation angles were estimated by the ergonomist, and the wrist position was characterized by extensive rotational movements, it is assumed that 2D angles tended to lead to a lower risk assessment by the ergonomist than that with data generated from MoCap. The application of 2D pose estimation can be performed quickly and practically with a mobile device or tablet, without requiring large financial resources. The paper-to-pencil RULA method is normally dependent on an ergonomist expert but is supported by angles through pose estimation that quantify body posture more precisely than the simple evaluation of the ergonomist. While the examination of a participant with the IMU-based MoCap took at least 20 min, an examination with pose estimation can be performed within 3 to 5 min, which can have a large impact on healthcare services in the corporate sector. However, the present study also showed that experienced ergonomists must be familiar with the application to ensure the correct interpretation of the results and the derivation of measures for action. Regarding data analysis, mean values were used, as opposed to extreme values, to average the stress in the work process over time. An analysis of extreme values would have added value and could be analyzed in future studies. Regarding sample acquisition, four left-handed participants were included in the experimental task. Therefore, the holding and guide arms were swapped to include both sides. Nevertheless, the change in arm guidance could lead to unnatural load patterns in untrained movement sequences, which do not occur in everyday working life. Unlike Holzgreve et al. [13], this study did not control for any further malformations of the spine, stiffed spinal joints, or rheumatic diseases that might have an influence on the participants’ experimental task. Nevertheless, no participant reported any spinal pathology. Further studies might include professional medical assessment of the spine and check for flexibility and control of MSDs via validated questionnaires.

### 4.3. Future Research

A larger and more diverse sample representing various workplace environments (e.g., office, manufacturing, and healthcare) would provide deeper insights into how the technologies perform across different settings, leading to a more comprehensive evaluation of their effectiveness in ergonomic assessments. The 2D pose estimation technology should be further improved to detect both body halves and deliver real-time RULA through machine-learning initiatives, such as in a previous study [17]. Both technologies should also be tested in combination with other observational methods, such as OWAS, REBA, [9] CUELA [42], or K-Score [43]. Unlike RULA, which focuses on joint angles and muscle use, OWAS considers the overall posture and load, potentially highlighting risks that RULA may not capture. This could lead to a broader understanding of postural risks and a more comprehensive assessment of the workers’ ergonomic conditions.

Additional images from both planes and time-continuous video-based joint angles over experimental tasks and longer work processes should be integrated into RULA by pose estimation to further increase the reliability of this technology and the detection of subtle movements, e.g., trunk and wrist. A technical solution for estimating transversal angles must be found to obtain an automated posture score. Furthermore, the paper-and-pencil RULA method should be incorporated into study designs to compare results from pose estimation and traditional observations [15]. In addition, male student participants should be included in future studies to determine potential sex-based differences in results. Finally, the potential influence of adding a dentist to the experimental task on the working process of the dental assistant students cannot be ruled out. A particular issue arises from the difference in height between the dentist and the dental assistant—typically, the dentist adjusts the treatment chair to an ergonomically favorable height, while the dental assistant remains standing. This should also be considered in future studies.

## 5. Conclusions

In summary, both technologies yielded similar overall RULA scores, with IMU-based MoCap showing greater variability and sensitivity to subtle postural changes. This technology tended to detect greater variability and higher postural deviations in the arms and wrist, whereas pose estimation appeared to emphasize trunk and neck deviations more strongly. Although IMU-based MoCap provided more detailed assessments of body parts, pose estimation was faster and more cost-effective; however, it was limited regarding occlusions and depth perception. The advantage of IMU-based MoCap is its tracking of continuous movement for a more accurate representation of ergonomic risks, especially in real-world variable work conditions.

## Figures and Tables

**Figure 1 bioengineering-12-00403-f001:**
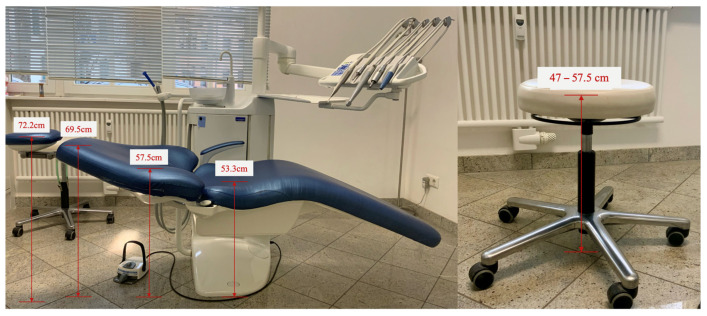
Dimensions of the patient positioning and treatment chair (specifications represent the total height of the patient and dentist assistant chairs).

**Figure 2 bioengineering-12-00403-f002:**
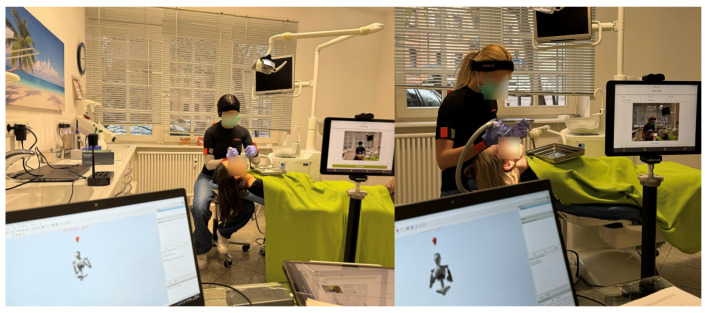
Test procedure and equipment. The Awinda Station (Movella) and Xsens software (Analyze Pro 2024.2; Xsens Technologies B.V.) are shown to the left of the photographs. Two-dimensional pose estimation via a tablet in the frontal plane is shown to the right of the photographs.

**Figure 3 bioengineering-12-00403-f003:**
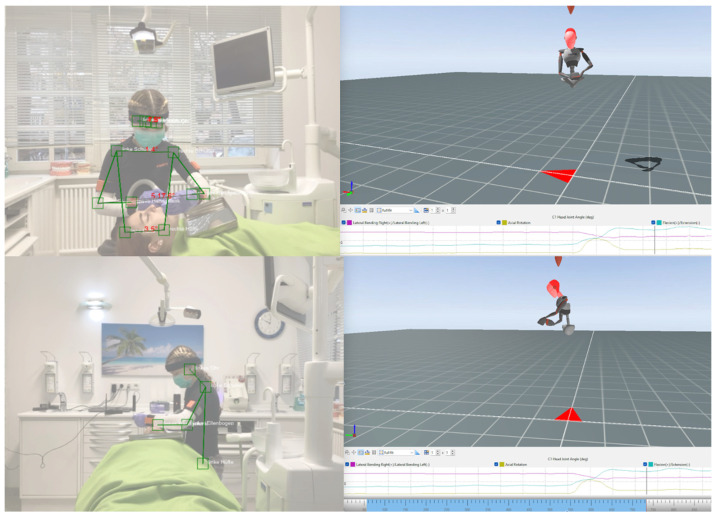
Pose estimation (left; frontal plane [upper photograph] and sagittal plane [lower photograph]) and inertial measurement unit (IMU)-based motion capture (right) in Xsens software (MVN Analyze Pro 2024.2; Xsens Technologies B.V. Enschede, The Netherlands). N.B.: The joint angles in pose estimation (left) were additionally displayed in an overview table. Green lines and boxes represent joint detection and resulting angles (ears, eyes, shoulders, elbows, wrists, and hips). Kinematic IMU data (right) are shown below the image.

**Figure 4 bioengineering-12-00403-f004:**
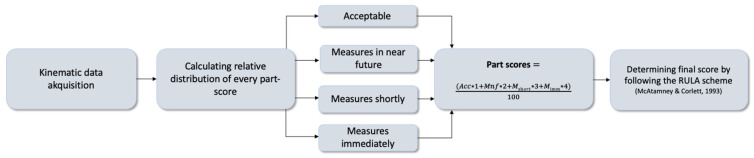
Data processing for calculating RULA scores from kinematic data collected by an inertial measurement unit-based motion capture.

**Figure 5 bioengineering-12-00403-f005:**
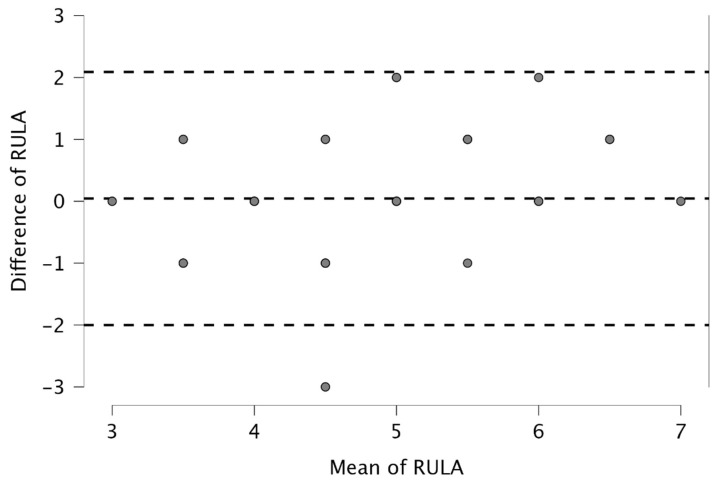
Bland–Altman plot of RULA scores assessed by pose estimation and inertial measurement unit-based motion capture. Overlapping points are not displayed. Dotted lines = standard deviation × 1.96 from mean value (upper and lower lines).

**Figure 6 bioengineering-12-00403-f006:**
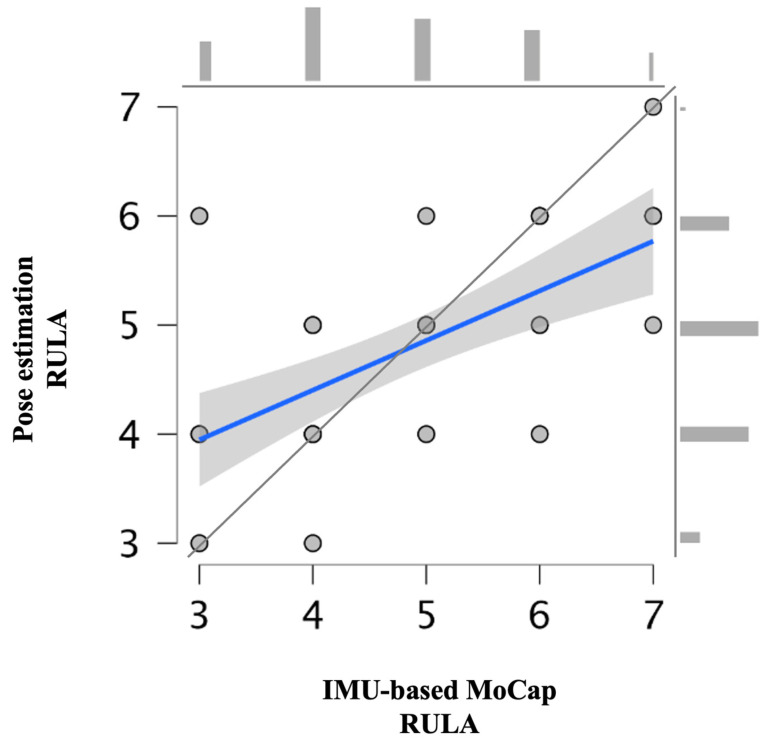
Scatter plot and histograms (above and right) representing overall RULA scores from pose estimation and inertial measurement unit-based motion capture including data trend line (blue) and 45° line (gray).

**Figure 7 bioengineering-12-00403-f007:**
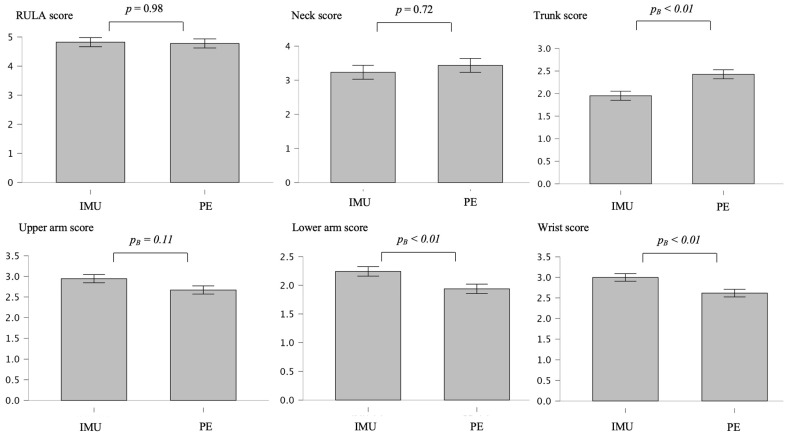
Mean differences and normalized error bars representing 95% confidence intervals regarding overall and body part RULA scores assessed by pose estimation (PE) and inertial measurement unit (IMU)-based motion. *p_B_* = Bonferroni-corrected *p*-value.

**Table 1 bioengineering-12-00403-t001:** Anthropometric data of the participants. Units: age: years, weight: kilograms, height(s): centimeters, width and span(s): centimeters.

	Age (y)	Weight	BMI	Height	Shoulder Height	Shoulder Width	Elbow Span	Wrist Span	Arm Span
Mean	19.56	63.41	21.56	165.00	138.74	37.97	81.42	126.95	162.79
SD	5.91	13.87	4.63	6.35	5.42	2.75	4.27	11.38	7.86
Min	15.00	42.20	14.08	152.20	125.50	29.80	72.80	110.50	145.00
Max	42.00	115.00	33.74	178.40	150.00	44.40	89.20	187.20	183.80

BMI, body mass index; SD, standard deviation.

**Table 2 bioengineering-12-00403-t002:** Steps to calculate RULA from pose estimation angles based on images from two planes (sagittal and frontal).

Parameters	Modifications of RULA
Posture of the upper arm	Sagittal view: angle between shoulder–hip line and shoulder–elbow axis
Shoulder raising	Frontal view: application shows the inclination of the shoulder; +1 when one shoulder was raised
Upper arm abduction	Frontal view: angle between hip–shoulder line and shoulder–elbow axis
Arm supported	Application showed whether the participants were supporting themselves or not (−1)
Posture of the lower arm	Frontal and sagittal view: angle between shoulder–elbow line and elbow–wrist axis
Arm working across midline or out to side of body	+1 as soon as the wrist went beyond the center of the body
Wrist posture	Frontal and sagittal view: assessment of the angle between the elbow–wrist line and alignment of the hand
Wrist bending from midline	+1 as soon as the fingers were not an extension of the ellbow–wrist line
Rotation of the forearm or hand	Rotation of forearm or hand was scored with +1 or +2 depending on hand posture
Muscle use score of arm and wrist	Static and dynamic muscle use was consistently scored as +1
Force/load score	This score was fixed to 0 because there was no lifting of dental instruments >2 kg in the dental practice
Posture of the neck	Sagittal view: angle between shoulder–hip line and shoulder–ear axis
Neck twist	Frontal and sagittal view: subjective assessment of the deviating position of the eye and nose from the body–midline
Neck tilt	Frontal view: +1 as soon as the angle of the eye line is over 0°
Trunk tilt	Sagittal view: subjective assessment of the alignment of the shoulder line to the hip line
Trunk twist	Frontal and sagittal view: subjective assessment of the alignment of the shoulder line to the hip line
Legs and feet supported	The value was fixed to +1 because the dental assistants remained seated during treatment, and their legs and feet were supported
Muscle use score of neck, trunk, and legs	Static and dynamic muscle use was consistently scored as +1
Force/load score	This score was fixed to 0 because there was no lifting of dental instruments >2 kg in the dental practice

**Table 3 bioengineering-12-00403-t003:** Descriptive results of pose estimation and inertial measurement unit-based motion capture assessments of overall and body part RULA scores.

	IMU RULA	PE RULA	IMU UA	PE UA	IMU LA	PE LA	IMU Wrist	PE Wrist	IMU Neck	PE Neck	IMU Trunk	PE Trunk
Mean	4.82	4.78	2.94	2.67	2.24	1.94	3.00	2.62	3.23	3.43	1.95	2.43
Median	5.00	5.00	3.00	2.75	2.50	2.00	3.02	2.75	3.10	3.50	2.00	2.50
SD	1.25	0.97	0.54	0.52	0.44	0.30	0.49	0.43	1.32	0.75	0.39	0.54
Min	3.00	3.00	1.50	1.50	1.00	1.25	1.75	1.50	1.00	1.50	1.00	1.00
Max	7.00	7.00	3.97	3.75	2.99	2.50	3.98	3.50	5.00	5.00	3.02	3.75

IMU, inertial measurement unit; PE, pose estimation; RULA, rapid upper limb assessment; UA, upper arm; LA, lower arm.

## Data Availability

The original contributions presented in this study are included in the article/Appendix A. Further inquiries can be directed to the corresponding author.

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
