# Peer review of "2D Pose Estimation vs. Inertial Measurement Unit-Based Motion Capture in Ergonomics: Assessing Postural Risk in Dental Assistants"

_bioengineering, 2025, doi:10.3390/bioengineering12040403_

Round 1

Reviewer 1 Report

Comments and Suggestions for Authors

Recommendation: Major Revision Required

The manuscript titled "2D Pose Estimation vs. Inertial Measurement Unit-based Motion Capture in Ergonomics: Assessing Postural Risk in Dental Assistants" presents valuable insights into the comparison of two methodologies for assessing postural risk in dental assistants. While the study contributes to the field of ergonomics and presents a novel comparison of technologies, substantial revisions are needed to improve the clarity, thoroughness, and overall quality of the manuscript before it can be considered for publication. The authors are kindly requested to revise the manuscript by addressing the following key areas in greater detail to enhance the rigor and depth of the work:

  1. How do the methodological differences between IMU-based MoCap and 2D pose estimation affect the reproducibility of postural risk assessments in ergonomic evaluations?
  2. Explain the rationale behind using Cohen’s weighted kappa for evaluating the agreement between the two technologies and how does it influence the interpretation of results?
  3. What are the potential implications of the higher variability (standard deviation) observed in the IMU scores compared to 2D pose estimation, and how does this affect the reliability of postural risk assessment?
  4. In your study, the mean RULA scores for both technologies are relatively close. What are the implications of this similarity for ergonomic assessment practices in industries with resource constraints?
  5. Considering the lack of significant agreement in body part scores such as upper arm, wrist, and neck, how do you propose improving the accuracy of these measurements in 2D pose estimation systems?
  6. What factors could contribute to the differences in RULA scores between the two technologies for body parts like the trunk and wrist, and how might these factors be addressed in future research?
  7. Why do you think the IMU-based MoCap system showed a tendency for more conservative risk classification compared to 2D pose estimation?
  8. The Bland–Altman plot demonstrated a fluctuation range between the two technologies. How should practitioners interpret these differences in real-world ergonomic assessments?
  9. expand on the limitations of using only two perspectives (sagittal and frontal planes) in 2D pose estimation for ergonomic analysis? What other angles might enhance the assessment?
  10. The study excluded participants with recent injuries. How could including participants with various musculoskeletal conditions impact the results and relevance to a wider workforce?
  11. Why did you opt to use the MoveNet model for 2D pose estimation, and what are the limitations of this model in assessing the dynamic postures of dental assistants?
  12. What are the potential implications of the relatively low agreement between the technologies when assessing joint-level differences in posture, especially in clinical practice?
  13. In the context of ergonomics, how do you propose balancing the trade-off between the time-efficiency of 2D pose estimation and the precision of IMU-based MoCap?
  14. Could the differences in RULA scores observed for the neck, wrist, and trunk be related to the subjective nature of scoring in the RULA method itself? How could this be addressed in future ergonomic assessments?
  15. What improvements could be made to the ergonomist’s training in pose estimation analysis to increase the reliability and validity of the results?
  16. Given the significant differences in RULA scores between the two technologies for the lower arm and trunk, how should this affect the ergonomic interventions or adjustments recommended for dental assistants?
  17. The study used a relatively small sample size of 45 participants. How might the inclusion of a more diverse and larger sample affect the outcomes of this comparison study?
  18. How would the results of this study change if different types of ergonomic assessments, such as the Ovako Work Posture Assessment System, were included for comparison?

Author Response

Dear Reviewer,
First of all, we want to thank you very much for taking your valuable time to review this manuscript. We have understood all comments as constructive suggestions for improving and have incorporated the aspects more critically in our manuscript. We hope to have addressed all questions and hopefully met the expectations. Please find the detailed responses below and the corresponding revisions in track changes in the re-submitted manuscript.

Reviewer 2 Report

Comments and Suggestions for Authors

This is an interesting study, which compared the measurement technologies of IMU-based MoCap and two-dimensional (2D) pose estimation for the assessment of ergonomic risk in female dental assistant students (n=45; mean age = 19.6 years) using the RULA method. A 30-second IMU-based MoCap and image-based pose estimation in the sagittal and frontal planes were performed during a representative experimental task, whereas data were analyzed using Cohen’s weighted kappa and Bland–Altman plots. The results demonstrated a significant moderate agreement between RULA score from IMU-based MoCap and pose estimation, and no significant agreement regarding the part-scores upper arm, lower arm, wrist, neck, and trunk. The Authors concluded that both technologies yielded similar overall RULA scores, with IMU-based MoCap showing greater variability and sensitivity to subtle postural changes. They suggested that future research should focus on refining video-based pose estimation for realtime postural risk assessment in the workplace.

The manuscript is generally well written. However, the design of this paper should be little improved before publishing.

I suggest:

  1. To add information regarding %right-handed participants in 2. Materials and Methods, 2.1 Participants (Page 3, Line 5).  
  2. To describe more detail the exclusion criteria of participants, that may affect the results of this study (2. Materials and Methods, 2.1 Participants (Page 3). How was controlled that the participants did not have: (1)  Rheumatism,(2)  Spinal deformities,(3)  Pain, incl. low back pain?   
  3. To delete the 1st sentence in 5. Conclusions (Page 12; Lines 445-446) „This study compared IMU-based MoCap and 2D pose estimation using MoveNet for student dental assistant ergonomic risk assessments using the RULA method“, because this is repetition of the aim and not associated the main results of this study. 

Author Response

Dear Reviewer,
First of all, we want to thank you very much for taking your valuable time to review this manuscript. Please find the detailed responses below and the corresponding revisions in track changes in the re-submitted manuscript.

Round 2

Reviewer 1 Report

Comments and Suggestions for Authors

 Accept in present form

Comments on the Quality of English Language

 Accept in present form